# Strengthening Medical Care for Young People in the Netherlands: A Reflection

**DOI:** 10.3390/ijerph191811487

**Published:** 2022-09-13

**Authors:** Danielle Jansen, Yvonne Vanneste-van Zandvoort, Károly Illy, Arne Popma, Marjolein Y. Berger

**Affiliations:** 1Department of General Practice & Elderly Care Medicine, University Medical Center Groningen, University of Groningen, 9713 GZ Groningen, The Netherlands; 2Department of Sociology and Interuniversity Centre for Social Science Theory and Methodology (ICS), University of Groningen, 9712 TG Groningen, The Netherlands; 3Accare, University Centre for Child and Adolescent Psychiatry, 9723 HE Groningen, The Netherlands; 4Dutch Knowledge Centre for Youth Health, Churchilllaan 11, 3527 GV Utrecht, The Netherlands; 5Department of Pediatrics, Hospital Rivierenland, 4002 WP Tiel, The Netherlands; 6Dutch Paediatric Society, 3528 BL Utrecht, The Netherlands; 7Department of Child and Adolescent Psychiatry & Psychosocial Care, Amsterdam UMC, 1105 AZ Amsterdam, The Netherlands; 8Amsterdam Public Health Research Institute, 1105 AZ Amsterdam, The Netherlands

**Keywords:** medical care, children, adolescents, the Netherlands, cooperation, coordination, health care professionals

## Abstract

To improve medical care for young people in the Netherlands, various professional groups representing physicians who provide medical care to children have developed a vision called ‘strengthening medical care for young people’. The purpose of this viewpoint is to reflect on the implementation of proposals to augment cooperation and coordination between the professional groups involved. Our reflection demonstrates that additional action regarding cooperation and coordination is still necessary to strengthen this care for young people. First, regarding the practical implementation of collaboration, the guidelines are unclear, and many are out-of-date. Second, adequate structured interdisciplinary training and intervision are lacking for physicians frequently collaborating in the care of young people. Third, interdisciplinary access to patient files is too complex and time-consuming. We recommend structured monitoring of the implementation of all improvement proposals, regarding both processes and outcomes. In addition, we recommend collaboration with physicians treating mentally disabled individuals to improve medical care for this group.

## 1. Introduction

Although most young Dutch people grow up rather healthy, happy, and vigorous, this is not true of every child. The most recent Health Behaviour in School-aged Children (HBSC) study demonstrates that especially young people in pre-vocational secondary education, young people with a migration background, young people from families with low incomes, and young people growing up with only one parent, are at greater risk of, for example, smoking, bullying and problematic social media use [1]. These results are not unique to the Netherlands; in general, adolescents in Europe and North America growing up in lower-income families experience poorer health and well-being [2]. Dutch research indicates that in addition to these greater risks of unhealthy and risky behaviour, some children have unmet care needs, especially in terms of mental health problems [3,4].

In response to these observations, and with the intention to improve medical care for young people, various professional groups representing physicians who provide medical care to children decided to develop a vision to improve medical care for this group, titled: ‘strengthening medical care for young people’ [5]. The aim of this vision was to answer the question: ‘What should physicians do to guarantee longer-term adequate, coherent medical care for young people?’ To answer this question, general practitioners, child and youth healthcare physicians, pediatricians, child and youth psychiatrists, their management representatives, a knowledge centre (Dutch Centre for Youth Health), and a parent organisation (The Parents Turn) united to form a steering committee [5].

First, the steering group critically examined the functioning of the medical profession, and indicated from their own daily practice the bottlenecks they experienced in medical care for young people:Insufficient communication and exchange of information. According to the steering group, referral, consultation, and data exchange were inadequate and insufficiently ensured. The group experienced a lack of communication and awareness of the need for cooperation among the various professional groups. Physicians providing medical care to young people made insufficient use of each other’s expertise. No profession could oversee the whole; the responsibilities for the various aspects of care were often not explicitly assigned and communicated to each other, or to the patient or his carers.Insufficient signalling and prevention. According to the steering group, early identification and prevention by physicians of unhealthy lifestyle, obesity, alcohol use, and psychological problems among young people functioned poorly and were insufficiently coordinated. Physicians failed to make effective use of each other’s expertise, possibilities, and data (e.g., health monitoring data).Gaps in the care of psychological and psychiatric problems. The steering group identified the following problems: (1) long waiting times for young people referred to specialist mental healthcare; (2) lack of easy-to-access mental healthcare services; (3) lack of a possibility to request a one-time psychiatric consultation, partly because health insurers pay only for diagnoses requiring specialist treatment; (4) overly long treatment of children by child and adolescent psychiatrists, partly as a result of the above; (5) fragmented child and adolescent mental healthcare, sometimes taking place outside medical care (e.g., care provided by social workers); (6) insufficient coordination or cooperation between child and adolescent mental healthcare and adult mental healthcare; (7) inadequate response by physicians to young people with behavioral problems.Not always the right care at the right place for the presented problem. The steering group believed that walls between professions and organizations render it difficult to realize connection between medical and non-medical care. Insufficient communication and consultation took place between the various parties and sharing information and/or consultation with the GP was often overlooked. The steering group also pointed to inadequate awareness of the possible connection between parenting, growing up, welfare problems, and medical problems. This awareness was considered important for all those involved in guiding young persons on their way to the right form of care.Insufficient knowledge and skills of healthcare professionals. According to the steering group, physicians often have an image of children that is formed and supported by their own experiences. Their image of the child will be determined by their own position in care and the morbidity pattern they observe. For example, a GP may score the wellbeing of a child as high because the child hardly visits his/her practitioner (apart from minor ailments). The GP is less aware of problems at school, and that will bias his/her interpretation of the child’s wellbeing. As a result, because children’s problems are sometimes not properly or only partially identified, they do not receive the care they need.Difficulty or failure in reaching certain risk groups. The steering group reported a lack of appropriate methods and instruments for identifying and defining risk groups (e.g., children of parents with psychiatric or addiction problems) between the ages of 4 and 19. They also pointed to a lack of structure to subsequently offer these young people (preventive) healthcare, and to insufficient knowledge about techniques to motivate parents and young people to adhere to health-promoting behaviors.

Second, based on these bottlenecks, the steering group formulated its vision: “A young person who visits a physician with a possible problem is helped without delay and without detours by the designated medical professional(s), in the right way and in the right place, preferably in mutual cooperation. Medical professionals are alert to both medical and non-medical questions/complaints from the young person and his parents/carers and explicitly take into account their possibilities and limitations.” This vision was accompanied by concrete proposals as to how medical care for young people could be improved:(1)Increasing the knowledge and skills of medical professionals.(2)Improving risk factor detection, identification, and prevention.(3)Realization of more cohesion, cooperation, exchange, and division of responsibility.(4)Enabling young people and their carers to obtain access to information and take part in remote interaction.(5)Promoting appropriate care and optimal medication policy.

Now, almost 10 years after the appearance of this vision, it is time to take stock. Our aim here is: (1) to reflect on implementation of the concrete improvement proposals laid down in the Dutch vision: ‘strengthening medical care for young people’, and (2) to determine whether we can rest on our laurels or must undertake further action to achieve the previously established goals. In this viewpoint we focus on realizing the proposals aimed at improving cooperation and coordination between the professional groups involved. This focus was chosen due to the large returns to be gained. Research highlights that better collaboration and coordination between specialties and primary care, using collaborative care models, leads to improved mental health outcomes for children and adolescents [6,7]. In addition, effective communication and improvements in teamwork have been associated with both delivery of better chronic care and positive health outcomes, like improved depression and anxiety outcomes [8,9]. Our viewpoint is not explicitly an evaluation, but an inventory, which reflects on and discusses the results of the vision.

## 2. A Reflection on the Proposals for Improving Cooperation and Coordination between Healthcare Professionals in Medical Care for Young People

The first proposals for improving cooperation and coordination focus on increasing knowledge of each other’s expertise and skills. Three concrete proposals were presented: (1) The relevant mono- and multidisciplinary guidelines from the various specialisms needed multidisciplinary coordination; (2) The care standards for children with complex (medical and other) problems needed to be explored; (3) Local interdisciplinary training and intervision for physicians (and possibly other care providers) who frequently collaborate in the care of young people had to be realized.

### 2.1. Multidisciplinary Coordination of Medical Guidelines from the Various Specialisms

Now, in 2022, a large number of guidelines are available for medical care for children. In addition, national collaboration agreements have been formulated for frequently occurring complex problems. Almost every medical profession has its own database with guidelines, but most of these have been written from a monodisciplinary perspective. Multidisciplinary guidelines include additions, adapted for specific needs in the daily practice of each profession. For example, the multidisciplinary guideline “asthma in children” [10] can be found in the database of the Federation of Medical Specialists [11], but is also accompanied by a guideline exclusively for child and youth healthcare physicians involved in the prevention of asthma [12]. An accompanying guideline for GPs focuses mainly on the treatment of asthma in primary care [13]. Finally, the guideline for pediatricians focuses on the treatment of asthma in secondary care [14]. However, for childhood asthma, an additional guideline with recommendations for collaboration does not exist (see below).

All professional associations have on their websites documents that explain the procedure for developing a guideline. The Federation of Medical Specialists (under which the guidelines of the Dutch Psychiatric Association are incorporated), The Society for Child and Youth Healthcare Physicians, and the Dutch General Practitioner Association all indicate in their documents that representatives of adjacent professional groups not directly involved in the development process have been consulted [15,16,17]. Although the document of the Dutch Pediatric Society [17] does not explicitly describe cooperation with other relevant professional groups in the development process, the procedure followed by the Federation of Medical Specialists (which includes the guidelines of the Dutch Pediatric Society) makes clear that cooperation is possible if desired.

In addition to the medical guidelines, since 2011 collaboration agreements have been jointly compiled by the Child and Youth Healthcare Physicians and GPs: the so-called National Primary Care Collaboration Agreements (NPCC; abbreviation in Dutch: LESA), followed by the National Transmural Collaboration Agreements (NTCA; abbreviation in Dutch: LTA). These agreements describe the process of collaboration and, where appropriate, make recommendations for working arrangements between child and youth healthcare physicians, GPs, pediatricians, and child and youth psychiatrists. However, many of these agreements are outdated and no longer fit with the current organization of care. 

### 2.2. Exploration of the Need for a Care Standard for Children with Complex (Medical and Other) Problems

To the best of our knowledge, no inquiry has been made regarding the need for care standards for children with complex problems. Such standards help healthcare professionals to organize care and support for a certain group of children in the best possible way; for example, for children with a chronic condition [18] and for children receiving medical childcare at home [19]. However, because no needs assessment has been carried out, care standards may be developed ad hoc and fail to meet the true needs.

### 2.3. Realization of Local Interdisciplinary Training and Intervision for Physicians (and Possibly Other Care Providers) Who Frequently Collaborate in the Care of Young People

Intervision is an organized consultation between colleagues to improve functioning [20] by discussing a child’s or adolescent’s health status, condition, or treatment. Interdisciplinary training or intervision is not discussed in the policy documents of the various professional associations. This does not, however, mean that no interdisciplinary training or intervision takes place, but it takes place incidentally and not on a structural basis. One interesting example of interdisciplinary training is the annual conference of Dutch pediatricians, where one of the three days is always organized in collaboration with another professional group (e.g., child and youth psychiatrists or child and youth healthcare physicians). In addition, more and more local, transmural, and interdisciplinary partnerships are emerging; for example, between child and youth healthcare physicians and pediatricians who share knowledge and get to know one other during theme evenings [21]. Finally, incidental interdisciplinary intervision is performed to discuss certain themes, like excessive crying, somatically unexplained physical disorders, and prenatal care. However, the inventory shows that there is no structured offer of interdisciplinary training or intervision, and GPs are often not involved.

The second series of improvement proposals of the steering group for improving cooperation and coordination was related to cohesion, exchange, and division of responsibility. Two concrete proposals were presented to serve as a solution for the bottlenecks concerning cooperation and coordination: (1) Facilitate electronic exchange of patient data and access to personal records managed by parents/carers; (2) Facilitate simple and fast consultation options.

### 2.4. Facilitation of the Electronic Exchange of Patient Data and Access to Personal Records to Be Managed by Parents/Carers

The reflection below concerns facilitating of two types of data exchange [2]: interdisciplinary data exchange (between different professional groups) and data exchange via (digital) files of parents.

As of 1 July 2017, the new Client Rights Act for electronic data processing was phased in. As part of this new Act, since 2020 clients (16 years and older) have been asked for ‘specified consent’: they must be able to indicate which data may or may not be viewed, and by which (categories of) care providers. This always concerns care providers with whom a client currently has a treatment relationship. Parents must give permission for data exchange between care providers for children up to the age of 16. According to the national government [22], the law makes it easier for healthcare providers to view patient data. However, an evaluation to confirm this assumption has not yet taken place.

Since 1 July 2020 physicians are obliged to provide, free of charge, an electronic inspection and an electronic copy of the medical file at the patient’s request. However, it is not mandatory to offer an online patient portal where the patient can view his or her file at all times. According to the law, parents are allowed to view the medical file of children up to 12 years old; if the child is 16 years old, the parent is not entitled to inspect the medical file unless the child gives permission [23].

To date, in the Netherlands little research has been done to assess the added value of combining information from electronic health records from different professional groups. A recent study, combining electronic health records from Dutch GPs and child and youth healthcare professionals with the aim to improve the prediction of mental health problems, concluded that when compared with the use of GP data alone, the use of this combined information did not improve the prediction of mental health problems [24].

### 2.5. Simple and Fast Consultation Options, including Their Financing, for GPs and Child and Youth Healthcare Physicians or Pediatricians, and Child and Adolescent Psychiatrists

Although it is not yet a national policy, more and more local initiatives are providing for a pediatric consultation hour in general practice to allow children to meet the GP and pediatrician together. Financing of these local initiatives seems to cause no problems, as this new way of working appears to yield significant savings for the health insurer (e.g., less use of expensive hospital equipment).

## 3. Conclusions

Our reflection on the implementation of concrete improvement proposals regarding cooperation and coordination as laid down in the Dutch vision ‘strengthening medical care for young people’ shows that we cannot yet rest on our laurels. Additional action regarding cooperation and coordination is still necessary to strengthen this care. First, our reflection shows that the development of multidisciplinary guidelines seems to be well aligned and coordinated among the different professional associations. However, it is crucial to ensure that this intention is implemented in practice: Is the level of participation in the development of guidelines by the various professional groups satisfactory and useful? In addition, how are the implementations of the National Primary Care Collaboration Agreements and National Transmural Collaboration Agreements progressing? Another important point of attention is monitoring the actuality of, specifically, the National Primary Care Collaboration Agreements and the National Transmural Collaboration Agreements. The organization of Dutch health and youth care is constantly subject to change, calling in turn for changes in the collaboration between the care providers involved. This requires frequent and timely updates of collaboration agreements.

Second, structured interdisciplinary training and intervision is largely lacking for physicians who frequently collaborate in the care of young people. Although the literature on (the effectiveness of) interprofessional knowledge transfer is limited, it is plausible to assume that such transfer can enable professionals to better understand and reach adjacent professional groups.

Third, although legislation regarding patient file access by professionals is continuously adapted to the wishes of patients and care providers, in this area in the Netherlands is still much room for improvement. Under current legislation, the physician is allowed to view patient files from another practitioner only if the patient has given explicit permission. Thus, the physician is forced to obtain from and grant permission for each person. Arranging this is very time-consuming, all the more so because it is not always clear to a GP, for example, which child and youth healthcare physician or pediatrician should be approached for permission for an individual patient.

To conclude, medical care for children in the Netherlands is often offered in a fragmented way, by medical specialists in various echelons. Lack of knowledge of each other’s expertise and skills, and the presence of certain rules and legislation, hinder well-coordinated care for children. This does not benefit the core values of health care: accessibility and continuity. With the vision ‘strengthening medical care for young people,’ the joint medical professions have attempted to improve this. We emphasize here that it is necessary to evaluate the implementation of these improvement proposals to prevent such initiatives from dying a quiet death. We recommend involving the physician for mentally disabled individuals in future actions. Due to the deinstitutionalization and socialization of care in the Netherlands, more and more young patients with intellectual disabilities are receiving their medical care through the child and adolescent healthcare physician or the GP. Collaboration with the physician for mentally disabled persons can improve medical care for this specific group.

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
