# Peer review of "Strengthening Medical Care for Young People in the Netherlands: A Reflection"

_ijerph, 2022, doi:10.3390/ijerph191811487_

Round 1

Reviewer 1 Report

The reading of the content it is not easy to follow, it lacks continuity and there are many concepts that repeat constantly.

It is necessary to improve the grammar, it is not easy to understand the text in general.

I recommend to research more about the topic, collecting data and information from other countries like Netherlands to contrast the methods proposed in the paper, and support them with statistics.

Increase evidence that interdisciplinary actions could improve children´s and teenagers mental and general health.

Author Response

Dear reviewer, thank you very much for your valuable feedback. We have made substantial changes to the manuscript on a number of points. We sincerely hope that our manuscript meets with your approval.

Best wishes,
Danielle Jansen

Reviewer 2 Report

I have reviewed a paper by Jansen et al. entitled “Strengthening medical care for young people in the Nether- 2 lands: a reflection”. Here are my comments regarding this manuscript:

-          Maybe the keywords can be reviewed

-          Line 39 – what do you mean by saying “little health”

-          Line 41 – explain “problematic social media use”

-          Line 42 – what category of children?

-          Line 51 – is the word “practising” necessary? – if so, please correct it

-          Lines 129-135 – this seems like a critical accusation – malpractice may I say – maybe you can rewrite not to sound so accusatory

-          Lines 136-138 – it appears that a verb is missing here

-          Line 149 – please explain “intervision”

-          Lines 158-160 – “In case of a multidisciplinary guideline, this guideline is found 158 in the database of the medical professions involved, often accompanied by a medical profession specific guideline” – please rewrite

-          Lines 155-166 – used guideline/guidelines for 10 times

-          Line 184 – what do you mean by out-of-date? Regarding what issue?

-          Line 186 – maybe you can explain what a “care standard” should be about; I find the next paragraph confusing

-          Line 187 – “no needs assessment has been carried out into the need” – please rewrite

-          Line 187-192 – you have used the word standards for 4 time, in each phrase

-          Most part of the conclusion section should be placed in the discussion because it comprise a series of ideas about the problems related to the main issue of this manuscript, the care for young people

-          The conclusion section should be redone

Author Response

(The authors gave the same response as above.)
